# Cerebral microcirculation mapped by echo particle tracking velocimetry quantifies the intracranial pressure and detects ischemia

Zeng Zhang[1,5], Misun Hwang[2,3,5], Todd J. Kilbaugh[4], Anush Sridharan[2] & Joseph Katz [1✉]

Affecting 1.1‰ of infants, hydrocephalus involves abnormal accumulation of cerebrospinal fluid, resulting in elevated intracranial pressure (ICP). It is the leading cause for brain surgery in newborns, often causing long-term neurologic disabilities or even death. Since conventional invasive ICP monitoring is risky, early neurosurgical interventions could benefit from noninvasive techniques. Here we use clinical contrast-enhanced ultrasound (CEUS) imaging and intravascular microbubble tracking algorithms to map the cerebral blood flow in hydrocephalic pediatric porcine models. Regional microvascular perfusions are quantified by the cerebral microcirculation (CMC) parameter, which accounts for the concentration of micro-vessels and flow velocity in them. Combining CMC with hemodynamic parameters yields functional relationships between cortical micro-perfusion and ICP, with correlation coefficients exceeding 0.85. For cerebral ischemia cases, the nondimensionalized cortical micro-perfusion decreases by an order of magnitude when ICP exceeds 50% of the MAP. These findings suggest that CEUS-based CMC measurement is a plausible noninvasive method for assessing the ICP and detecting ischemia.

[1] Department of Mechanical Engineering, Johns Hopkins University, Baltimore, MD, USA. [2] Department of Radiology, Children's Hospital of Philadelphia, Philadelphia, PA, USA. [3] Department of Radiology, Perelman School of Medicine, University of Pennsylvania, Philadelphia, PA, USA. [4] Department of Anesthesiology and Critical Care Medicine, Children's Hospital of Philadelphia, Philadelphia, PA, USA. [5]These authors contributed equally: Zeng Zhang, Misun Hwang. ✉email: katz@jhu.edu

Hydrocephalus involves abnormal cerebrospinal fluid (CSF) accumulation in the brain ventricles[1]. Affecting 1.1‰ of infants[2], it is the leading cause for brain surgery in newborns and results in long-term neurologic disabilities in up to 78% of patients[3,4]. Intracranial hypertension and ventriculomegaly are major complications of hydrocephalus[3,5], which could lead to compromised cerebral blood flow (CBF), brain ischemia, inflammation, secondary neurovascular damage, and brain herniation[3,6]. Aimed at alleviating the elevated intracranial pressure (ICP), the most common treatment consists of CSF diversion to another anatomical location outside of the central nervous system via an invasive ventricular shunt[5]. The clinical decision for this surgical intervention relies primarily on computed tomography (CT) depicting ventriculomegaly following clinical symptomatology[1,5]. Unfortunately, by the time of radiographic diagnosis of ventriculomegaly, the ICP is often elevated significantly, and the brain tissue is at risk of ischemia and irreversible brain damage. Compounding the difficulty of early diagnosis and intervention, medical or surgical, is the fact that some patients have an elevated ICP without ventricular enlargement[7–9]. Therefore, a brain CT is often not sensitive enough to detect an elevated ICP and ongoing ischemia, placing the infant at risk for permanent brain damage. Hence, there is a dire need for other techniques that could guide timely treatment.

Since accurate ICP measurements utilizing invasive sensors are rarely adopted due to the risks associated with infection, catheter tract hemorrhage, parenchymal damage, and neurological deficits[10], considerable efforts have been invested in developing noninvasive methods[11]. They include transcranial Doppler[12], MRI[13], near-infrared spectroscopy (NIRS)[14], diffuse correlation spectroscopy[15], and venous ophthalmodynamometry[16]. These clinical studies have shown that hydrocephalus is associated with increased ICP[16], reduced CBF[12,13,17], reduced cerebral oxygen saturation, and increased oxygen extraction[18]. However, prior studies have shown that the global CBF[13] or flow indices in macro-vessels[19] are inadequate for monitoring the ICP. In contrast, several in vivo results have shown that capillary rarefaction and significantly declined capillary flows are associated with elevated ICP[20,21]. It has also been reported that when cerebral perfusion pressure (CPP) is reduced, the percentage changes to microvascular flow in several cerebral regions, including the bregma, striatum, cortex, and cortical rim, are significantly higher than the changes to the total CBF[22]. Accordingly, histology and microscopy in animal studies and autopsies have demonstrated associations among hydrocephalus, increased ICP, decreased capillary density within the cortical parenchyma, and microvascular injury[23–26]. Furthermore, animal studies have shown that the increased capillary resistance due to an elevated ICP causes non-nutritive thoroughfare channel shunt flow, which leads to tissue hypoxia[20]. These findings suggest that a parameter that accounts for the cerebral micro-perfusion, including the density of micro-vessels and the velocity in them, could be used as a potential marker for ICP and ischemia associated with hydrocephalus. Implementing such a method requires techniques capable of mapping the microvascular perfusion in various cerebral regions, which could be applied in clinical settings.

In recent years, contrast-enhanced ultrasound (CEUS) imaging, which utilizes intravascular microbubbles (<5 μm in diameter) with high echogenicity to visualize blood flow, has been introduced for evaluating cerebral perfusion by monitoring the temporal evolution of the intensity in neonatal brain CEUS images[27]. In the ultrasound community, ultrasound localization microscopy (ULM), which is also referred to as super-resolution ultrasound, has been developed to generate high spatial resolution maps of blood vessels using many localized bubble positions[28,29]. Separately, several postprocessing techniques have been developed in the fluids community to quantify

the flow parameters based on CEUS images, including ultrasound imaging velocimetry or echo-PIV[30–32], and echo-particle tracking velocimetry[33,34]. Combining bubble tracking with super-resolution ultrasound imaging has enabled mapping of the flow velocity in blood vessels[35–37]. Pioneering in vivo works include the demonstration of isolated microbubble tracing in tumor vessels[35], and the super-resolution flow mapping in a mouse ear[36]. Subsequently, this method has been used for reconstructing the cerebral and renal vascular systems in rodent models[38–40], as well as for detecting a small deep-seated human cerebral aneurysm[41]. Early applications have used pre-clinical/clinical CEUS scanners with framerates typically <100 frames per second[29,35,36,42], and recent studies have already utilized ultrafast ultrasound imaging, extending to more than 1000 frames per second[38,41]. Further details can be found in a recent review article[43]. This technique demonstrates great promise for noninvasive bedside assessment of cerebrovascular physiology. Hence, in this paper, it is implemented in a high-fidelity porcine model of neonatal hydrocephalus, which has close approximation to human anatomy and cerebrovascular response[44], to quantify the relationship between ICP and the spatio-temporal distribution of cerebral blood flow.

Using a clinical CEUS system and a particle tracking velocimetry (PTV) method, we map the micro- and macro- circulations in a coronal cerebral plane of hydrocephalic pediatric pig models at varying ICP levels. Eight animals are involved, where five of them are used as the original cohort for developing the functional relationships between ICP and cerebral blood flow, and the other three serve as the validation cohort. The data are used for visualizing and quantifying the changes to the spatial distribution of perfusion in different parts of the brain, including the cortex and the thalamus. Quantitative analysis of perfusion is performed using the time-averaged velocity in the macro-vessels, and a cerebral microcirculation (CMC) parameter for the regional microvascular perfusion, which accounts for the concentration of micro-vessels and the velocity in them. Combining the regional or global CMCs with hemodynamic parameters leads to two highly correlated functional relationships with the ICP. Furthermore, when the lactate to pyruvate (L/P) ratio obtained from cerebral microdialysis exceeds the ischemic threshold, there is a significant decrease in the cortical CMC. These findings suggest that the CEUS-based CMC measurements could potentially serve as a noninvasive tool for evaluating the ICP level and detecting brain ischemia in neonatal hydrocephalus.

## Results

**Visualization of the cerebral vasculature.** Figure 1a is a sample visualization of the vascular distributions across the entire hemispherical coronal plane (piglet #0121) at baseline ICP. The corresponding distribution of time-averaged velocity magnitude is presented in Fig. 1b. The sample magnified vasculature and velocity distributions for several locations (Fig. 1c) demonstrate that blood vessels that are separated by ~120 μm can be readily distinguished. Moreover, sample time-averaged velocity profiles (Fig. 1c) have parabolic profiles with peak magnitude of 2.2 cm/s and 3.4 cm/s at the center of a 250 μm vein, and a 390 μm artery, respectively. The arteries are readily identified by the flow pulsatility during the cardiac cycle. The venal values are consistent with previously published data[45]. The present arterial velocity for MAP = 68 mmHg is lower than simulated values[45] which are based on a steady 100 mmHg intravascular pressure at the entrance to a 2 mm artery, as expected. To facilitate the analysis of regional perfusion differences in response to ICP changes, several areas are selected and labeled in Fig. 1c. Included are a macro blood vessel (region #5), the thalamus (region #1), and the cortex divided into three sub-regions (regions #2–4) based on

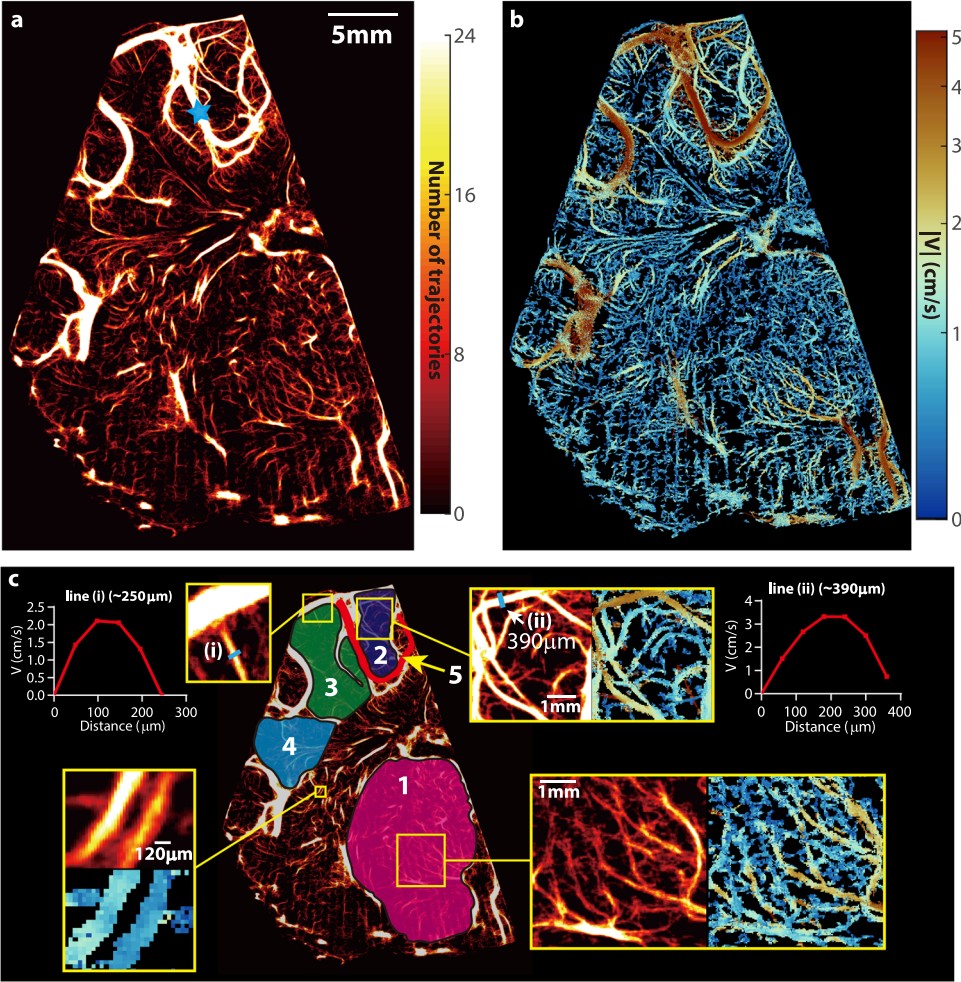

**Fig. 1 Cerebral vascular map and velocity distribution for piglet #0121 at the baseline ICP. a** A heatmap of all the trajectories containing at least four exposures visualizing the micro- and macro-vascular distributions in a coronal plane. The blue star marks the location of pulsed-wave Doppler ultrasound measurement. **b** The corresponding time-averaged velocity distribution. **c** Several sub-regions are labeled for statistical analysis of perfusion, including the micro-vessels in the thalamus (#1), several parts of the cortex (#2, #3, and #4), as well as a macro blood vessel (#5). A few regions are magnified to provide a closer view of the current regions of interest. Sample velocity profiles across two different blood vessels are also provided in (**c**). One measurement is made for this sample case, where 5760 images are used to generate results.

their locations relative to macro-vessels. For thalamus and cortex, taking advantage of the high spatial resolution features of the vascular map, the areas occupied by macro-vessels with in-plane width more than 400 μm (corresponding to the second-order branches of the middle cerebral artery in piglets[46]) are excluded (Fig. 1c, maximum width is ~390 μm), and the analyses are focused on the micro-vessels. Initially, these microvascular regions are labeled manually. Subsequently, an automated procedure (Supplementary Methods) based on the vesselness filter[47] has been implemented for defining the same microvascular regions of interest, by detecting and excluding vessels with diameters equal or larger than 400 μm. The flow parameters (discussed below) obtained for the cortical regions #2–4 using the two methods are statistically identical (Supplementary Fig. S1). Hence, the current microvascular flow evaluations are not affected by the values of major vessels associated with region selections. Only macro vessel #5 is used since it is the only one that appears consistently for all piglets, presumably owing to slight variations in the location of the imaging plane. A visual demonstration of the changes to the vascular map and velocity distribution caused by elevated ICP for the same piglet is provided in Fig. 2a. In the cortex, the number of micro-vessels with

blood flow and the velocity in each vessel gradually decrease as the ICP increases, with a particularly noticeable reduction from 40 to 46 mmHg. The same trends occur in macro vessel #5. In contrast, the changes in the thalamus appear to be relatively small up to an ICP of 30 mmHg. However, once the pressure is increased to 40 and then to 46 mmHg, there is a reduction in microcirculation in the thalamus as well. Quantifications of these trends are discussed in the following section.

**Quantification of the impact of elevated ICP on the CBF.** Two parameters have been adopted as surrogates for the cerebral perfusion: (i) The CMC parameter is calculated by summing the time-averaged velocity (in mm/s) in each pixel of the area occupied by micro-vessels (<400 μm), and dividing the result by the total area of these regions (i.e., all pixels, including those with zero values). Hence, the CMC accounts for the changes to the density of micro-vessels and the velocity in them due to elevated ICP. Note that CMC values are inherently lower than the spatio-temporal averaged velocity since the CMC accounts for areas without vessels. (ii) The time-averaged velocity magnitude, $V_5$ (mm/s), is utilized for characterizing the flow in the macro blood

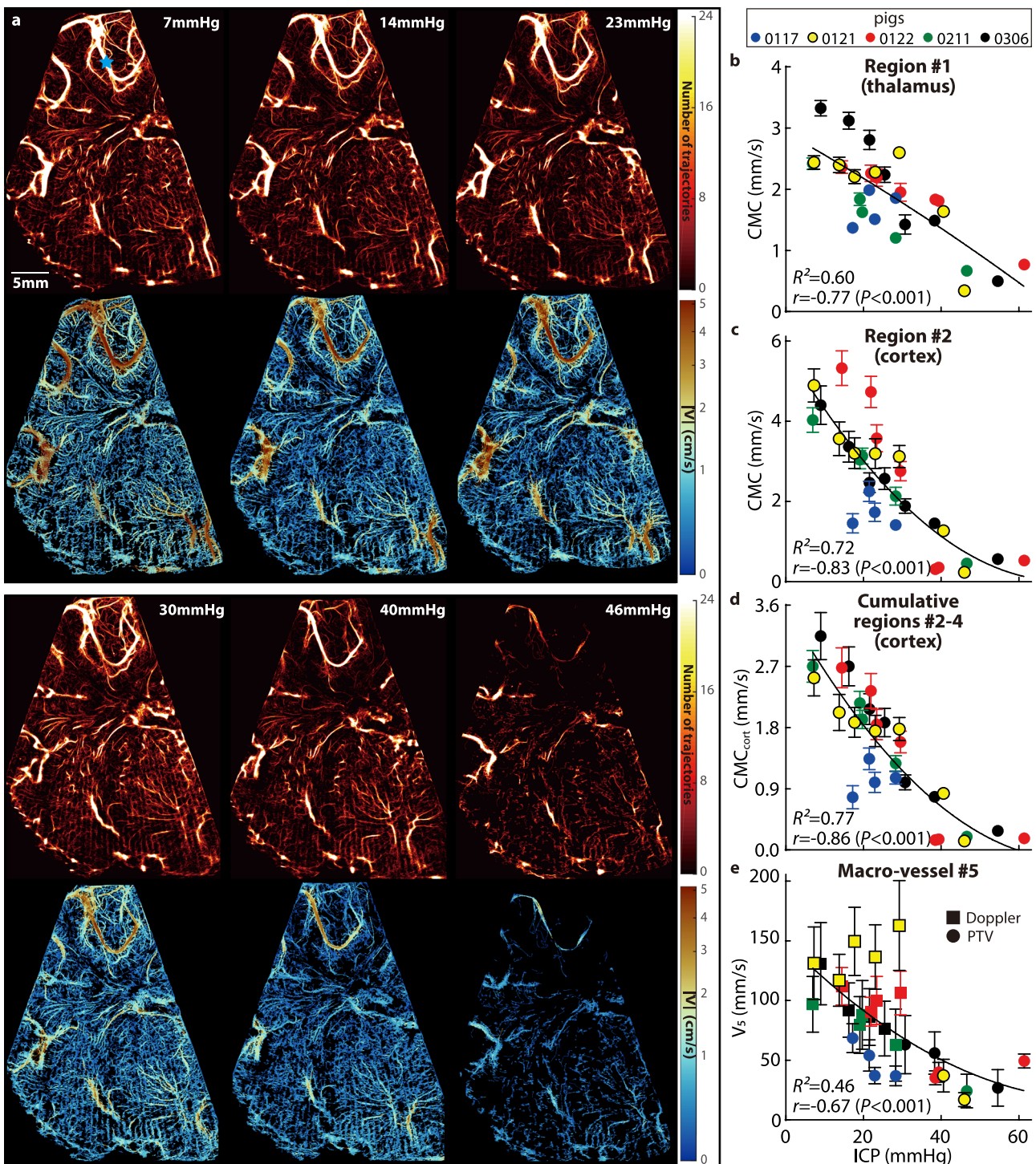

**Fig. 2 Visualization and quantification of the influence of increasing ICP on the cerebral micro- and macro-perfusion. a** Sample visualizations of cerebral blood vessels (top row) and the corresponding velocity distributions (bottom row) for piglet #0121 demonstrating the effect of increasing ICP on the cerebral perfusion. **b**–**d** Relationships between the ICP and the cerebral microcirculation (CMC, original cohort) parameters in the thalamus (**b**), in the cortical subregion #2 (**c**), and in all three cortical subregions combined (**d**, CMC_cort). **e** Relationships between the ICP and the time-averaged velocity ($V_5$) in the macro blood vessel #5 at the point marked with a blue star in (**a**). For (**b**–**e**) data are presented as mean ± SD, where the error bars show the temporal standard deviations of CMCs or $V_5$ over $n = 20$ cardiac phases, several standard deviation values for high ICP are too small to be shown. The curves are least-square fitted quadratic functions, with the corresponding coefficient of determination ($R^2$) shown on each plot. Also presented are the correlation coefficients ($r$) and the corresponding $P$ values between the CMCs or $V_5$ and the ICP. Source data for (**b**–**e**) are provided as a Source Data file.

**Table 1 Test cases and baseline information for the present pediatric porcine models.**

|  | Piglet Number | Weight (kg) | Baseline | | | Tested ICP Levels (mmHg) | Microdialysis |
|---|---|---|---|---|---|---|---|
|  |  |  | MAP (mmHg) | PP (mmHg) | ICP (mmHg) |  |  |
| Original cohort | #0117 | 10.30 | 63.76 | 20.70 | 8.08 | 8[a],17,21,23,28 | No |
|  | #0121 | 10.10 | 68.44 | 15.62 | 7.26 | 7,14,18,23,29,41,46[b] | Yes |
|  | #0122 | 10.80 | 83.17 | 28.70 | 14.41 | 14,22,23,30,38,39,61 | Yes |
|  | #0211 | 11.20 | 65.81 | 20.64 | 6.97 | 7,19,20,28,47 | Yes |
|  | #0306 | 9.50 | 62.93 | 24.23 | 9.02 | 9,16,22,25,31,38,55 | Yes |
| Validation cohort | #0212 | 11.90 | 105.59 | 35.01 | 6.78 | 7,14,21,33,42,60 | Yes |
|  | #0219 | 10.50 | 98.22 | 26.31 | 2.65 | 3,13,21,30,34,47,57 | Yes |
|  | #0713 | 10.20 | 74.90 | 23.90 | 9.40 | 9,21,28,35,39 | Yes |

[a]Not included owing to inconsistency in the imaging plane.
[b]Microdialysis data unavailable owing to unstable physiological condition.
Hemodynamic data for elevated ICP levels are provided in Fig. S2a. Source data are provided as a Source Data file.

vessel #5. Owing to the limitations in the maximum velocity that can be measured based on CEUS data acquired at 48 frames/s (11 cm/s, see the "Methods" section), $V_5$ is based on the PTV data when all instantaneous speeds over the cardiac cycle are lower than 11 cm/s, and pulsed-wave Doppler ultrasound data recorded at the same location (Fig. 2a, blue star) are used for cases with higher speeds. The CMCs for all the piglets of the original cohort (Table 1) in the thalamus (Fig. 2b), cortical subregion #2 (Fig. 2c), cumulative cortical CMC combining regions #2, 3, and 4 ($CMC_{cort}$, Fig. 2d), and $V_5$ (Fig. 2e) decrease with increasing ICP. The error bars in Fig. 2b–e represent the temporal standard deviation of CMCs or $V_5$ for each case. The magnitudes of the (negative) correlation coefficients between ICP and CMCs are larger than 0.77 ($P < 0.001$), and that between the ICP and $V_5$ is 0.67 ($P < 0.001$). The coefficient of determination ($R^2$) for the shown least-square parabolic fits fall in the 0.60 to 0.77 range for the CMCs, and 0.46 for the $V_5$.

Aiming to improve the collapse of the data, hence obtaining higher $R^2$ for better evaluation of ICP levels, effects of hemodynamic parameters, which are expected to affect the cerebral perfusion, should also be accounted for in calculations. Based on the data of the original cohort (Table 1), plots of ICP and $CMC_{cort}$ vs. the mean arterial pressure (MAP), cerebral perfusion pressure (CPP = MAP-ICP), systolic and diastolic pressures (SBP and DBP), and pulse pressure (PP = SBP-DBP) are presented in Supplemental Fig. S2a, b, all showing weak or poor correlations. Yet, plots with combinations of parameters, namely MAP/ICP, PP/ICP, and (PP-ICP)/MAP appear to have stronger correlations (Fig. S2c, d). Accordingly, Fig. 3a summarizes the highest values of $R^2$ obtained by least-square fitting several functions to the CMC (or $V_5$) vs. hemodynamic or combined parameters for each cerebral region, confirming that these combined variables yield higher $R^2$. Subsequently, isolating ICP from these relationships leads to CMC/MAP, CMC/PP, and PP − (0.17CMC − 0.29)MAP (the constants 0.17 and 0.29 are based on the curve fit in Supplementary Fig. S2d), as likely candidates for obtaining collapsed trends. As Fig. 3b demonstrates, the latter two fits vs. ICP have significantly higher $R^2$ ($P < 0.01$ and $P < 0.001$ respectively) compared to the CMC vs. ICP relationship. Values of $R^2$ exceeding 0.8 are obtained for the cortical microvascular regions, while lower values are obtained for the thalamus or macro-vessel (Fig. 3b). For the data of the original cohort, plots of CMC/PP and $V_5$/PP vs. ICP are presented in Fig. 3c–i by colored symbols, each showing a parabolic curve fit and corresponding values of $R^2$. Owing to its highest $R^2$ (0.87) and correlation ($r = -0.90$, with $P < 0.001$), $CMC_{cort}$/PP = $5.2 \times 10^{-5}(ICP)^2 - 6.1 \times 10^{-3}(ICP) + 0.18$ could be considered as a potential functional relationship for evaluating the ICP (Fig. 3c). This parabolic function is tested by the validation data (Fig. 3c, hollow symbols). While the decrease in $CMC_{cort}$/PP persist, results are more

scattered, resulting in $R^2$ decreasing from 0.87 to 0.72 for all data (original + validation). The primary discrepancy occurs for ICP > 40 mmHg, where the majority of the original cases are ischemic, while only one of the validation cases involves ischemia (further discussion follows). Accordingly, the validation CMCs are higher than the original ones in this ICP range. A comparison of the relative errors in ICP prediction, i.e., the difference between the predicted and measured ICP level divided by the measured ICP, between the original and validation sets is presented in Fig. S3a. It shows that the relative validation errors for ICP = 10 (i.e., 10 ± 5) mmHg are higher than those of the original data, while there are no statistical differences between the original and validation data for ICP > 10 mmHg. The validation data indicates that the ICP can be predicted within 4.2 mmHg at low to moderate pressure (0 < ICP < 35), and within 13.8 mmHg for ICP > 40 mmHg. As for the macro-vessel, the scatter in the original data is caused primarily by piglet #0121. If the results for this pig are removed, $R^2$ for $V_5$/PP increases from 0.51 to 0.77. However, adding the validation cohort data reduces the $R^2$ to 0.08 for all data, and 0.01 for all data with pig #0121 excluded.

In exploring possible reasons for the differences in trends between the original and validation cohorts, especially at high ICP, we have found that the MAP for the validation cohort is generally higher than that of the original piglets by 25 mmHg (Fig. S2a). Hence, taking the effect of MAP into accounted has led to the following linear relationship (Fig. 3b and j):

$$ICP = PP - (0.17CMC_{cort} - 0.29)MAP + 0.08 \qquad (1)$$

This linear relationship has a $R^2$ of 0.91 for the original $CMC_{cort}$, and 0.84 for the combined original and validation cohorts. The corresponding relative error in ICP prediction between the original and validation cohorts are insignificant for all ICP levels. The absolute errors for the validation data vary from 3.1 mmHg for ICP = 0–35 mmHg to 7.8 mmHg for ICP > 40 mmHg. Hence, even with a validation cohort with higher MAP, the linear relationship presented in Eq. (1) provides a reasonable prediction of the ICP level. In contrast, while adding the effect of MAP makes the validation and original data of $V_5$ overlap (Fig. 3p), the results are scattered and $R^2$ for the combined data is only 0.12.

Note, the trends of the perfusion with increasing ICP differ among the cerebral regions. The rates of perfusion decrease are demonstrated by the nondimensionalized slopes of the fitted curves, namely, $k = [d(CMC/PP)/d(ICP)]_{average} \times (ICP \times PP/CMC)_{ref}$, where the subscript 'average' refers to the average slope in 8–40 mmHg ICP range, and 'ref' to values at ICP = 8 mmHg, i.e., the average baseline ICP for all pigs. Results for each region are presented in Fig. 3c–h. They indicate that rates of the perfusion reduction in the cortical regions are similar and higher than that in the thalamus (by 38%).

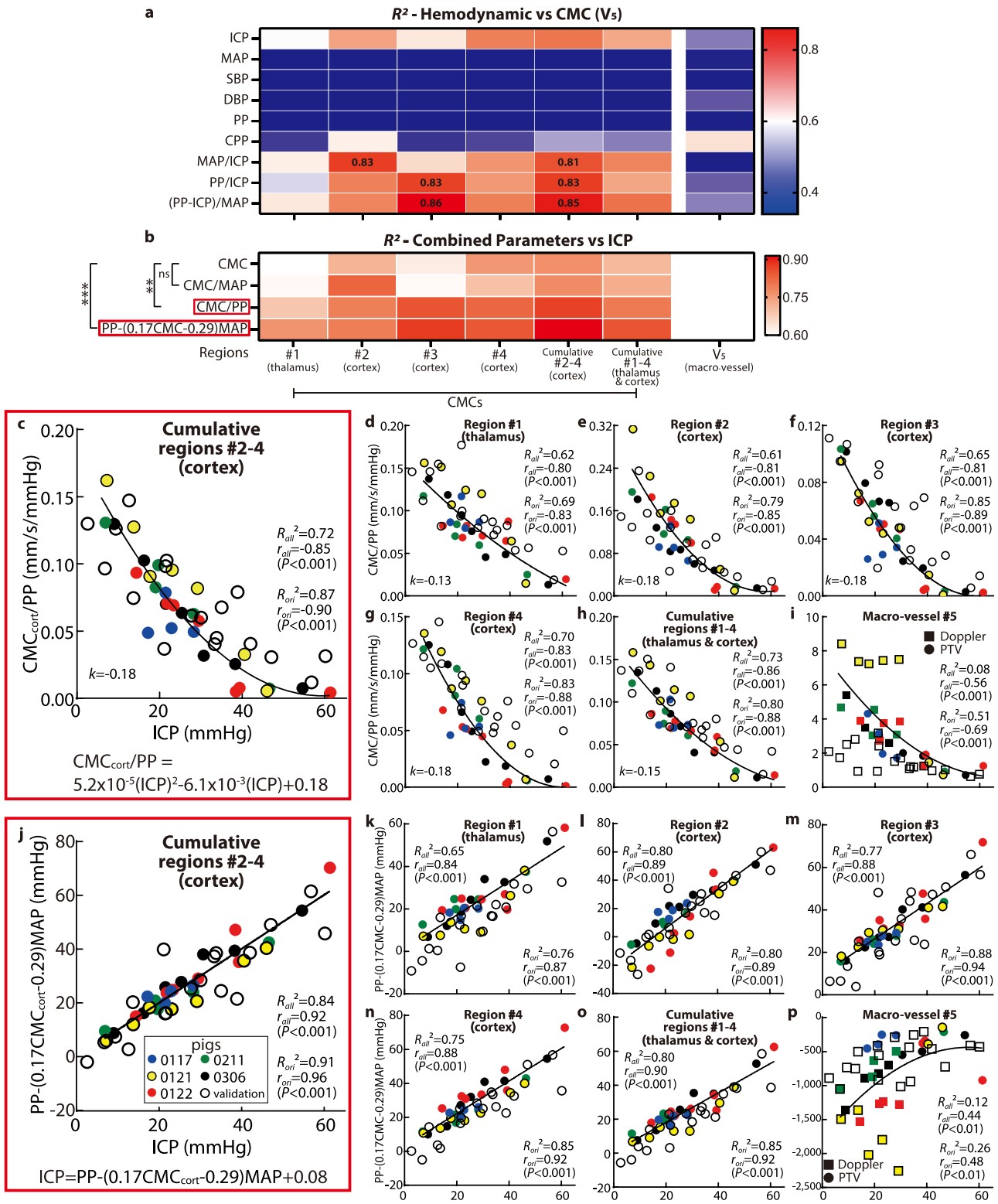

The cortical regions appear to be more vulnerable to intracranial hypertension.

**Microdialysis and CMC during ischemia**. Cerebral microdialysis data have been obtained for seven of the eight piglets (Table 1), and their relationships with ICP are presented in Fig. S4. As a characterization of the relationship between aerobic and anaerobic metabolism, a lactate to pyruvate ratio exceeding 30 has been linked to cerebral ischemia[48]. Figure 4a shows the variations of L/P with ICP for the current study. This trend is consistent with the results of a previous study[49] (Fig. 4a). The relationships between the cerebral perfusion and L/P are plotted in Fig. 4b. Here, from the perspective of fluid mechanics, the cortical microcirculation data are presented in a nondimensional form that compares the dynamic pressure of the blood flow to the pulse pressure, i.e., $\rho(\text{CMC}_{\text{cort}})^2/\text{PP}$, where $\rho$ is the average density of

**Fig. 3 Combining the perfusion parameters with hemodynamic variables to obtain highly correlated functional relationships for the assessment of ICP.**
**a** A heatmap of $R^2$ of the least-squared fits to the functional relationships between a series of hemodynamic parameters and CMC (or $V_5$) for the indicated cerebral regions (as defined in Fig. 1). **b** A heatmap of $R^2$ of the least-squared fits to the functional relationships between the ICP and the indicated combined hemodynamic-perfusion parameters in several cerebral regions. Here, two-way ANOVA multiple comparison results with Benjamini-Hochberg correction are also demonstrated, where 'ns' indicates $P = 0.72$, $**P = 0.0016$, and $***P = 0.0001$. **c–i** Variations of CMC/PP with ICP, in the: **c** combined cortical regions (#2-4, $CMC_{cort}$/PP); **d** thalamus (#1), and cortical subregions #2 (**e**), #3 (**f**), and #4 (**g**); **h** combined thalamus and cortical data (#1-4); and (**i**) macro vessel $V_5$/PP. Corresponding values of $R^2$ for the parabolic curve fit, as well as $r$ and $P$ for the two-tailed Pearson correlation are provided for each plot. **j–p** Trends of PP $-$ (0.17CMC $-$ 0.29)MAP with ICP in the: **j** combined cortical regions (#2-4, PP $-$ (0.17$CMC_{cort}$ $-$ 0.29)MAP); **k** thalamus (#1), and cortical subregions #2 (**l**), #3 (**m**), and #4 (**n**); **o** combined thalamus and cortical data (#1-4); and (**p**) macro vessel PP $-$ (0.17$V_5$ $-$ 0.29)MAP. Here, the corresponding $R^2$ for the fitting, as well as $r$ and $P$ for the two-tailed Pearson correlation are shown. For (**c–p**) the curve fittings are developed based on the original cohort (colored symbols), and validated by validation data (hollow symbols). Subscript 'ori' denotes the values based on the original cohort, while 'all' denotes the values based on original and validation cohorts combined. Source data for (**a–p**) are provided as a Source Data file.

blood. A corresponding comparison to the ICP non-dimensionalized by the MAP is presented in Fig. 4c, where the values of $\rho(CMC_{cort})^2$/PPfor original and validation cohorts collapse. Results of nondimensionalized perfusion parameters for other regions are provided as Supplemental Information (Fig. S5). A few trends are evident in Fig. 4a–c: (i) All the six cases (five original and one validation) with $\rho(CMC_{cort})^2$/PP $< 10^{-7}$, which deviate from the rest of the data for at least one order of magnitude, involve cerebral ischemia. Note that for all of them, ICP/MAP $\geq 0.5$. (ii) Among the other 37 cases, where $\rho(CMC_{cort})^2$/PP $> 10^{-7}$, 92% has L/P smaller than 30, i.e., they are considered non-ischemic. Hence, the non-dimensional cortical CMC could be used for detecting ischemia. Values smaller than $10^{-7}$ show 100% probability of ischemia while those larger than $10^{-7}$ indicate with 92% likelihood that they do not involve ischemia. In the macro vessel (Fig. 4d), even though the rate of perfusion reduction appears to be increased for both original and validation cohorts when ICP $\geq 0.5$MAP, their values are scattered. As a visualization of the reduction in perfusion during ischemia, Fig. 4e and f compare the vasculature maps as L/P increases from the baseline level to L/P $= 85$ for piglet #0306. In this sample, there is a 99.5% reduction in nondimensional cortical microcirculations and 97.3% in nondimensional $V_5$. Moreover, since ICP/MAP exceeds 0.5 for all of the ischemia cases, with similar MAP levels ($P = 0.25$) between ischemia and non-ischemia cases in this ICP range, the significant reduction in micro-perfusion might have been caused by the collapse and/or deformation of the micro-vessels owing to the elevated ICP.

**The impact of ICP on the cyclic variations of CMC.** Tracking of the bubbles also enables measurements of perfusion variations at different phases of the cardiac cycle. The phase-averaged CMCs (denoted as $CMC_\Phi$) are determined by dividing each cardiac cycle into 20 phases based on the ECG signal, and averaging the values corresponding to each phase over 250 cycles. Figure 5a shows $CMC_\Phi$ of each region for increasing ICP range. Results for all piglets are averaged, with the error bars showing the standard deviations among them. To focus on the cyclic variations, the peak value is subtracted from each data point, hence the results are denoted as $\Delta CMC_\Phi$. Figure 5b summarizes the trends of $\Delta CMC_\Phi$ by comparing the maximum values ($|\Delta CMC_\Phi|_{max}$). For the original cohort, in the cortical regions, $|\Delta CMC_\Phi|_{max}$ decreases gradually at varying rates as the ICP is increased up to 30 mmHg, and then drops sharply at higher pressures. Conversely, in the thalamus, where the cyclic variations are initially lower than the cortical levels, the cyclic variations appear to remain at a similar level up to 40 mmHg, and then decrease. The flow oscillation amplitudes for the validation cohort in these regions show the same trend (Fig. 5b), where they decrease with increasing ICP. However, since there is only one ischemia case for the validation cohort and

the MAP is generally higher, there are no sharp decreases when ICP $> 30$ mmHg. Further insights can be derived by non-dimensionalizing the cumulative cortical values (i.e., $\rho|\Delta CMC_{\Phi cort}|^2_{max}$/PP) and plotting them with ICP/MAPin Fig. 5c. Here, the cortical values for all piglets collapse and decrease at a growing rate with increasing ICP/MAP. For ICP/MAP $\geq 0.5$, the values for the ischemia cases in the original cohort (encircled in Fig. 5c) are significantly lower ($P < 0.01$) than those of the non-ischemia cases in the same ICP range, but that for the validation cohort is not. Since the magnitudes of PP for the ischemia cases, which drive the cyclic variations, are not different ($P = 0.19$) from those of the non-ischemia ones (Fig. S2), the reduced flow oscillation amplitudes indicate an increase in the blood flow resistance, consistent with the severe reduction in the mean micro-perfusion. The values of pulsatility index (PI $= |\Delta CMC_{\Phi cort}|_{max}/CMC_{cort}$) are presented in Fig. 5d, with baseline magnitudes ranging between 0.1 and 0.4, consistent with previous results for the cortical micro-vessels[50]. Moreover, the PI increases with ICP ($r = 0.48$, $P < 0.001$), in agreement with prior doppler based studies[51], but there is a considerable scatter in the data ($R^2 = 0.24$). The decreasing $|\Delta CMC_\Phi|_{max}$and increasing PI also indicate that the changes to the mean flow are stronger than the pulsatility when ICP increases.

## Discussion

Detailed maps of the CMC, obtained by tracking microbubbles in clinical CEUS images of hydrocephalic pediatric porcine models, are used for quantifying the impact of elevated ICP on the cerebral perfusion. The results show that the CMCs in the cortex and thalamus along with the velocity in a macro-vessel decrease with increasing ICP. Two functional relationships are obtained when the analysis incorporates hemodynamic effects. A decaying quadratic function ($r = -0.85$, $R^2 = 0.72$) describes the relationship between increasing ICP and $CMC_{cort}$/PP. Alternatively, PP $-$ (0.17$CMC_{cort}$ $-$ 0.29)MAP shows a highly correlated ($r = 0.92$, $R^2 = 0.84$) linear increase with ICP. In contrast, results based on the macrovascular perfusion or microcirculation in the thalamus are more scattered and less correlated. Hence, quantitative evaluation of the cortical microcirculation using the CEUS-based CMC parameter could be developed into a plausible method for noninvasive bedside assessment of ICP for neonatal hydrocephalus. Furthermore, when ICP/MAP $\geq 0.5$, several ischemia cases are identified by a lactate-to-pyruvate ratio exceeding 30. These cases involve significant reductions in the nondimensionalized cortical microcirculation ($\rho(CMC_{cort})^2$/PP) that deviate from the rest of the data. These findings suggest that measurements of cortical micro-perfusion could potentially be used for the detection of cerebral ischemia.

The present findings are qualitatively consistent with previous studies reporting a reduction in CBF in patients with hydrocephalus[13]. This reduction has been attributed to distortion of the periventricular vessels due to ventricular dilation[52] and

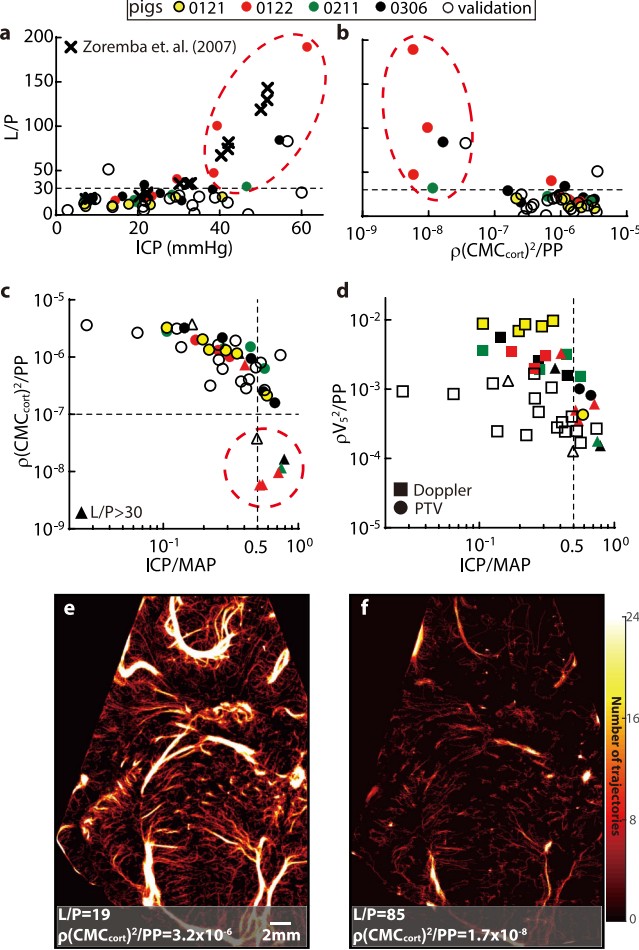

**Fig. 4 Comparison to microdialysis data showing a drastic reduction in perfusion once cerebral ischemia occurs. a** Trends of lactate-to-pyruvate ratio (L/P) with increasing ICP for the current experiment (circles) and previously reported results (cross[49]). The dashed line at $L/P = 30$ has been suggested as a threshold for cerebral ischemia[48]. **b** Trends of L/P vs. the nondimensionalized cumulative cortical micro-perfusion parameter $\rho(CMC_{cort})^2/PP$. **c** Trends of $\rho(CMC_{cort})^2/PP$ with ICP/MAP in logarithmic scales. In (**a**–**c**), the same group of encircled cases are identified to have cerebral ischemia and significantly low nondimensionalized perfusion, all occurring at $ICP/MAP \geq 0.5$. **d** Variations of $\rho V_5^2/PP$ with ICP/MAP. For (**a**–**d**), the colored markers show the original data; the hollow ones are validation data. Cases whose microdialysis data are not available are not plotted. For (**c**, **d**), cases whose L/P are larger than 30 are marked by triangles. **e**, **f** A sample comparison of the perfusion heatmaps for the baseline (**e**) and ischemia (**f**) cases for piglet #0306 showing the omnipresent sharp reduction in perfusion during ischemia. Source data for (**a**–**d**) are provided as a Source Data file.

increased flow resistance caused by compression of arterioles and venules[53]. Previously reported reductions in the number and caliber of capillaries in periventricular and cortical regions during hydrocephalus[52,54] also agree with the presently observed decrease in the number of micro-vessels. The trends of decrease in micro-perfusion with increasing ICP vary among brain sections. However, prediction of the ICP becomes more reliable (higher $R^2$) when the analysis is based on a broad area in the cortex.

Previous animal studies[55] have shown that when the CBF decreases below 20% of the normal level, the neurological damage to the brain becomes irreversible. In the current investigation, for the cases identified as ischemia, the values of $CMC_{cort}$ are <20%

of the baseline level. These findings suggest that the significant reduction in $\rho(CMC_{cort})^2/PP$ when $ICP/MAP \geq 0.5$ involves irreversible brain damage. The validity of this postulate needs to be verified by independent cerebral damage assessments. For humans, the reported threshold for irreversible tissue damage is 16–18% of the normal level[56]. Occurrence of ischemia is also accompanied with a significant reduction in the cyclic variations of perfusion during the cardiac cycle. While both PP and MAP for the ischemic and non-ischemic cases are similar when $ICP/MAP \geq 0.5$, the reduced mean and cyclic variations in flux for ischemic cases suggest that the contraction of the micro-vessels has reached a level that it severely restricts the flow in them. The suppressed micro-perfusion also appears to accelerate the decrease in macro-perfusion. Since the intravascular pressure decreases with increasing branching hierarchy, "remote" vessels (1) are likely to have lower mean flow and flow oscillation magnitudes[50,51], and (2) are more likely to be influenced by the external pressure. Accordingly, region 2, which appears to be located "closer" to major arteries, should have higher values of CMC and ΔCMC than regions 1, 3 and 4 at baseline level, in agreement with the present findings. Moreover, while the observed regional differences in flow pulsatility could be due to the aforementioned reasons, it may be dependent on other factors, including CSF dynamics and pulsatility, intrinsic regional differences in vascular morphology, compliance and regulation, and/or spatial heterogeneity in brain elasticity.

Quantifying the CMC appears to provide a feasible method for measuring the ICP and detecting ischemia. Hence, neurosurgical management of neonatal hydrocephalus could be based on early detection of elevated ICP and detailed assessment of brain ischemia. This approach has several advantages over other methods for characterizing the cerebral blood circulation. For example, analyses of the time-intensity variations of CEUS images using disruption & replenishment sequences or bolus injection[57] could be simpler alternatives for assessing the ICP levels. However, as the quantitative results show (Figs. S6–10 and Supplemental Method), all the parameters, except the rise time and max intensity & wash-in slope ratio of the bolus injection, are scattered without a clear trend with variations in ICP. The latter two increase significantly for ICP > 40 mmHg, but show no significant differences from the baseline level for the lower ICPs (Figs. S6–8). Moreover, for the ischemia cases, their results are very scattered and are not significantly different non-ischemic cases in the same ICP range. Hence, these parameters are not suitable for direct quantification of the ICP or for detecting ischemia. As for transcranial Doppler, the present data also show that trends of the velocity in the large vessel are less correlated with ICP than the microcirculation statistics. This finding is consistent with that of Hanlo et al.[19], namely that the Doppler indices measured in the major cerebral arteries are inadequate for monitoring the complex intracranial dynamic responses in patients with raised ICP. Next, MRI and CT have been used extensively for mapping the regional distributions of the CBF, but they cannot resolve the flow in specific blood vessels. The correlations between the MRI-measured CBF and ICP are low with substantial scatter in data[13]. Furthermore, CEUS imaging can be easily performed by the bedside. Another approach, namely NIRS, is limited to a shallow depth. Hence, it cannot quantify microcirculatory disturbances in the deep cortical matter, which has the most prognostic value in infants with brain injury[58]. In this context, the quantitative perfusion maps extend the potential applications of CEUS imaging well beyond the hydrocephalus population. For example, the risk of elevated ICP with ischemic brain injury is also high in infants with traumatic brain injury, stroke, in-hospital cardiac arrest, and during extracorporeal membrane oxygenation.

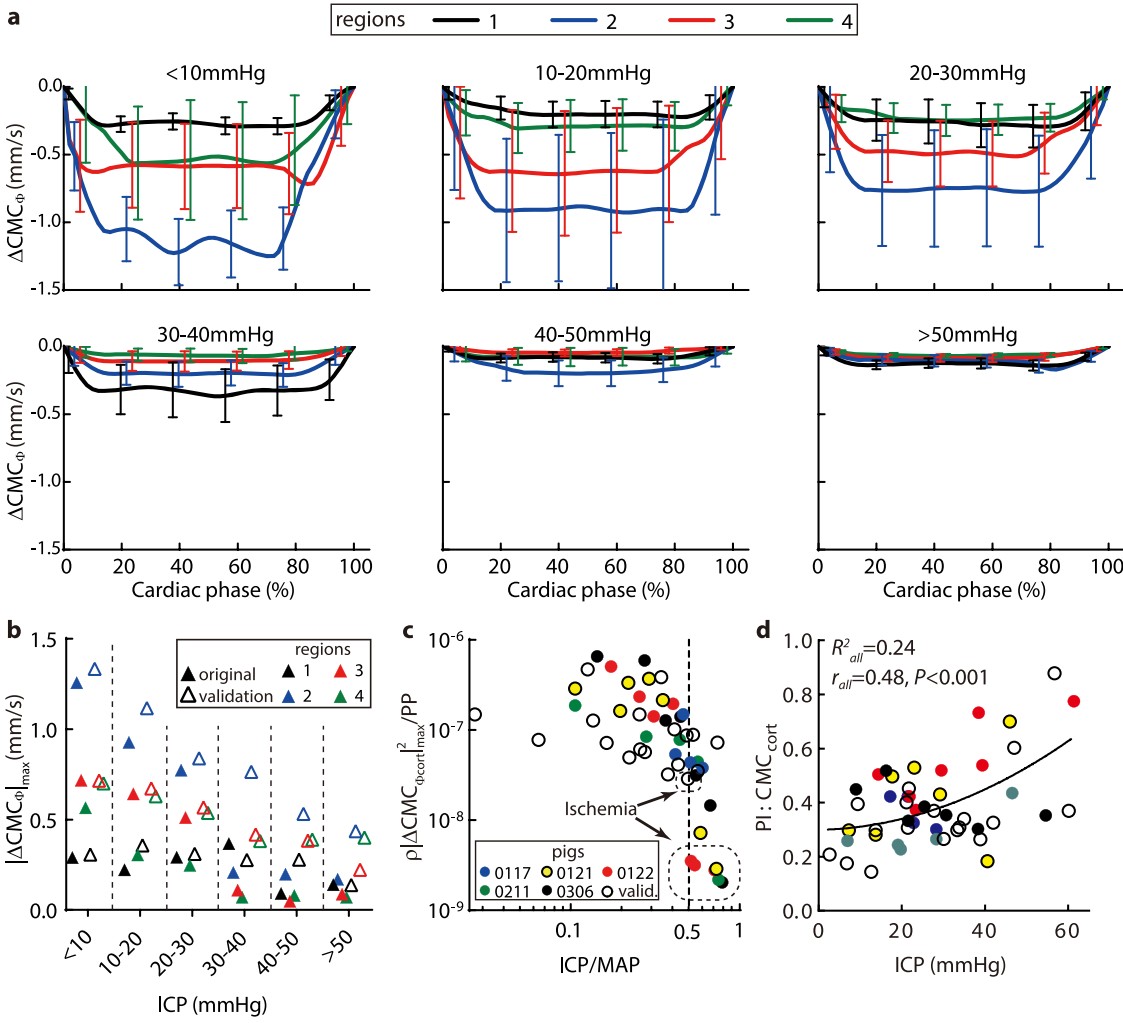

**Fig. 5 Influence of increasing ICP on the pulsatility of the micro-perfusion during the cardiac cycle. a** Variations with cardiac phase of the phase-averaged CMC (over 250 cardiac cycles) after subtracting the maximum (systolic) value ($\Delta CMC_\Phi$). Data are presented as mean ± SD, where error bars indicate the standard deviations among cases, there are $n = 3, 7, 11, 4, 3$, and 2 independent cases respectively for each increasing ICP level. **b** Trends of the maximum cyclic variations in micro-perfusion ($|\Delta CMC_\Phi|_{max}$) with increasing ICP for different brain regions. (**c**) The nondimensional cyclic variations in CMC for the cortex regions combined, $\rho|\Delta CMC_{\Phi cort}|^2_{max}/PP$, plotted vs. ICP/MAP. The encircles cases correspond to the ischemia cases marked in Fig. 4. (**d**) The pulsatility index for the $CMC_{cort}$. Corresponding values of $R^2$ for the parabolic curve fit, as well as $r$ and $P$ (0.0006) for the two-tailed Pearson correlation are provided. In (**b–d**) the colored markers indicate the original data while the hollow symbols indicate the validation data. Source data for (**a–d**) are provided as a Source Data file.

The present method has several limitations. First, the current pediatric pig model with cranial window is relevant for neonatal brain since the anterior fontanel is not closed, facilitating high-quality CEUS imaging[59]. The quality of data in adults is expected to be inferior due to the transcranial imaging. Hence, future efforts for adults or older children could attempt to obtain images through, e.g., the temporal bone[41] or eyes. Note that some grating-like artefacts are observed in the far field of the image, which are caused by the native ultrasound system using a curved-scanning ultrasound beam for generating images. However, the functional relationships developed in this work for quantifying ICP are not based on values in this region, hence the main claims are not affected by these artefacts. A detailed discussion about this artefact can be found in Baad et al.[60]. Second, due to the paucity of published clinical reports and safety data, the application of CEUS to diagnose intracranial pathologies is off-label in the United States[59]. For this reason, the present study doesn't have access to human data. Third, while the present measurements have been performed in a single cerebral plane, CEUS imaging

could be performed in multiple planes or extended to 3D imaging. Fourth, while the present findings demonstrate the potential role of cerebral micro-vessels as imaging biomarkers of ICP and brain ischemia, our study is limited to a small sample size and an acute hydrocephalus model. The cerebral micro-circulation and CMC vs. ICP relationship are likely to be affected by hemodynamics, medications, duration and severity of prior insult, maturation, genetics, metabolomics, and systemic factors. They might also be complicated by the phase of hydrocephalus, i.e., whether it is acute, subacute, chronic, or acute on chronic. For example, previous studies have reported contradictory results about the differences between acute and chronic phases in hydrocephalus. Rashid et al.[50] report that the mean cerebral capillary diameters of rats are 4.4 ± 0.7, 4.5 ± 0.8, and 4.7 ± 1.0 μm for the acute (5–7 days), chronic (3–5 weeks), and control animals, respectively. The corresponding mean blood flow rates are 0.6 ± 0.6, 0.8 ± 0.7, and 0.8 ± 0.9 nL/min, being higher during the chronic phase than that of the acute phase. In contrast, in Higashi et al.[61], the regional CBFs in the cortex, thalamus, and midbrain

show no differences between acute (~1 week) and chronic (4–6 weeks) phases in cat models, but both are lower than the control cases. In order to make our results generalizable to patients with hydrocephalus of various etiologies and disease duration, this study needs to be expanded significantly to investigate how the biomarker accuracy is affected by individual variabilities, accounting for all the aforementioned factors. Moreover, taking advantage of the high spatial resolution features of ULM results, future data analysis could probe into the microvascular structures well beyond the CMC parameter. For example, one could account for the impact of physiological conditions on several microvascular parameters, such as vessel sizes, branching orders, vessel types, individual microvascular velocities, etc. Accounting for these parameters is likely to improve the robustness of diagnostic potential of CEUS imaging in general, and assessment of ICP or ischemia in particular. Finally, clinical CEUS equipment has a limited framerate, typically <100 frames/s, restricting the range of velocities that can be measured by tracking bubbles. Using the current 48 frames/s instruments, the maximum measurable flow velocity is 11 cm/s. While this limit is sufficient for mapping the flow in the micro-vessels along with some of the arteries, it is inadequate for major arteries. This limitation could be resolved by recording data at higher framerates, which would also simplify the bubble tracking and allow higher bubble concentrations. The resulting shorter acquisition period would be more convenient for clinical applications. High framerate CEUS systems exceeding 1000 frames/s are already being used in research involving rodent models[38–40], and recently in a trial involving humans[41], but are not widely available for clinical applications at the present time.

## Methods

**Study design**. This study introduced a noninvasive method for estimating ICP and detecting cerebral ischemia based on the spatio-temporal distributions of cerebral blood flow. Using clinical CEUS imaging and ULM, we mapped the micro- and macro-circulations in a coronal cerebral plane of hydrocephalic piglets at varying ICP levels. This study was approved by the Institutional Animal Care and Use Committee of the Children's Hospital of Philadelphia. The original cohort for evaluating the ICP and perfusion consisted of five 4-week-old female piglets (10 ± 0.65 kg, Table 1), which had been identified as high-fidelity models for human children[44]. Additional three (10.8 ± 0.91 kg) were used for validation. Continuous invasive measurements included ICP, hemodynamic parameters, and cerebral microdialysis. For each piglet, the ICP was elevated from the baseline to several levels (Table 1) by infusing artificial CSF into the lateral ventricles via ventricular catheters until hemodynamic instabilities indicated impending brain herniation. For each level, the ICP was kept steady for 5 min before CEUS imaging, and collecting samples for cerebral microdialysis as biomarkers of brain ischemia. All the CEUS data were included in the analysis except for the baseline data for one of the piglets (Table 1), where the imaging plane was initially inconsistent with the rest of the measurements. We did not exclude any results as outliers.

**Animal preparation**. All piglets received anesthesia with intramuscular injection of ketamine (20–40 mg/kg) and xylazine (1.5–3 mg/kg), followed by inhaled 4% isoflurane and 100% FiO$_2$ via snout mask. After endotracheal intubation, mechanical ventilation was provided in 21% FiO$_2$ and 2% isoflurane to maintain anesthesia. For hemodynamic monitoring and intravenous fluid administration, the right femoral artery and bilateral femoral veins were cannulated under ultrasound guidance. The heart rate, respiratory rate, arterial blood pressure, ECG, and rectal temperature, were recorded for the duration of the experiments. The arterial blood pressure signals collected by an invasive hemodynamics monitoring sensor (Millar, USA) were compiled using the Blood Pressure Module of the Labchart software (ADInstruments, USA) to obtain the systolic and diastolic pressures. During the period of image acquisition, the systolic and diastolic pressures were stable with a maximum variation of ±3 mmHg. The analysis was based on the time-averaged hemodynamics values during the period of image acquisition. As for neuromonitoring, burr holes were prepared for the placement of a parenchymal ICP monitor (NEUROVENT PTO, Raumedic, Germany), a brain tissue oxygen monitor (Licox, Integra Life-Sciences, USA), and a cerebral microdialysis catheter (CMA 71 Elite, mDialysis, Sweden). The ICP probe was inserted through the right parietal bone, 1 cm from both the coronal and sagittal sutures, and 2 cm below the skull entering the subdural space. The Licox catheter system was placed via a frontal burr hole, terminating at the gray-white matter junction. The microdialysis catheter was placed through the right parietal bone and inserted 1 cm into the brain parenchyma. To simulate

hydrocephalus, a ventricular catheter was inserted into the left lateral ventricle for delivery of artificial CSF. It was infused at a rate needed for elevating the ICP to the desired level indicated by continuously ICP monitoring. The microdialysate samples were collected for each ICP level, and analyzed for pyruvate, lactate, glycerol, and glucose using an automated ISCUS Flex Microdialysis Analyzer (mDialysis, Sweden) and LABpilot software (mDialysis, Sweden). At the end of each experiment, the animal was euthanized using potassium chloride.

**CEUS imaging**. All of the CEUS scans were performed using the Siemens ACUSON Sequoia system (Siemens Medical Solutions, USA). To mimic the neonatal skull and obtain clear images, a 2 × 2 cm$^2$ cranial window was drilled upper right to the midline in the parietal region with intact dura. The window was selected to fit the tip of the 9EC4 transducer (Siemens Medical Solutions, USA), with the probe tip in contact with the dura. It should be noted that while the cranial window affected the amount of CSF required for reaching a certain ICP level, e.g., six times more CSF was needed for reaching ICP = 40 mmHg with the cranial window, the indicated ICP values are measured directly and reflects the actual pressure exerted on the brain tissue. The ultrasound probe was aligned in the coronal plane containing the maximum transverse diameter of the bilateral thalami and then fixed to the experimental table using a stereotactic arm. The piglet's head was also fixed to the table to minimize the motion caused by breathing. Due to the wide velocity range in this plane, over 10 cm/s in the large blood vessels and <1 cm/s in micro-vessels, it was necessary to acquire the images at the highest possible framerate. Hence, the ultrasound sector width was reduced to focus on one hemisphere, allowing image acquisition at a rate of 48 frames per second. For each level, the ICP was kept steady for 5 min to ensure that the hemodynamic conditions were stabilized prior to CEUS imaging. Then, a pulsed-wave Doppler measurement was conducted at the point indicated in Fig. 1a. Subsequently, a bolus of 0.2 mL Lumason (Bracco Diagnostics, USA) was prepared following the manufacturer's instructions, injected manually and followed by a quick flush of 10 mL saline. The image acquisition of bolus injection took 90 s. Then a veterinary syringe pump (PRACTIVET, USA) was connected to the femoral vein catheter to deliver the Lumason steadily at 0.6 mL/min for the bubble tracking procedures. The pump was gently shaken during the infusion to keep the microbubbles from sedimentation. When bubble concentration in the field of view appeared to stabilize (after 1 min of infusion), the image acquisition started. This infusion rate was selected, based on a pre-study, to establish a bubble concentration low enough to facilitate detection and tracking of individual bubbles while still maintaining a sufficient number for fully mapping the vascular structures. Ideal concentrations varied between 100 to 150 bubbles per image. For each ICP level, the CEUS cine clips were acquired for 120 s, each consisting of 5760 images. The grayscale tissue images were generated simultaneously while acquiring the CEUS images and used for correcting motion artifacts. Specifically, an affine transformation was used for matching the grayscale images in each series with the first one, and for repositioning the CEUS images accordingly. Subsequently, the field of view was cleared by bubble bursting for 5 s, and a disruption & replenish sequence was collected for another 30 s. To keep the bubbles fresh, a new vial of Lumason was used per ICP level. On average, it took 305 s on each ICP level for imaging and about 40 min to increase the ICP by 10 mmHg. Using the bolus injection and disruption & replenish sequences, the time-intensity based dynamic CEUS analysis were performed following[57,62]. The detailed procedures and results were presented as Supplemental Methods and figures (Fig. S6–10).

**Image processing and bubble tracking**. Data processing consisted of the following steps:

(i) Image enhancement aimed at localizing the bubble centers: As illustrated in Fig. 6a, the typical bubble trace diameter exceeded 10 pixels (600 μm), much larger than the actual size (~2.5 μm). The shape of these traces varied spatially, becoming increasingly elongated in the circumferential direction with increasing distance from the transducer. Therefore, the scattering pattern of each bubble was modeled as the convolution between a spatially varying point spread function (PSF) and an impulse signal. The PSF (Fig. 6b) was determined using modified blind deconvolution[31]. Specifically, the local PSF was estimated for each 80 × 80 interrogation window in 20 randomly selected images. An iterative procedure minimized the differences between raw images and convolutions of point sources with the candidate PSF, which was then averaged over the 20 images (Fig. 6b)[31]. The resulted PSFs were then used for deconvolving the traces, greatly reducing their sizes (Fig. 6c). The procedure was evaluated using synthetic images with imposed noise[31], showing 50% reduction in the uncertainty in the locations of bubble centers. Such deconvolution-based image enhancement was also used in other ULM studies[63,64]. Additional processing included subtraction of the non-uniform background, calculated by averaging 15 consecutive images, and enhancement using modified histogram equalization[34] (Fig. 6c).

(ii) Tracking of bubbles and mapping the microcirculation: Assuming that bubbles only traveled within blood vessels, heatmaps of bubble centers (Fig. 6d) could be used as guidances for likely vessel locations. Hence, overlap with this heatmap was one of the criteria for detecting bubble trajectories. As illustrated in Fig. 6e, tracking of a newly detected bubble at time $t_0$ (red circles) was initiated by searching for all the candidates displaced by less than a prescribed maximum distance in the following

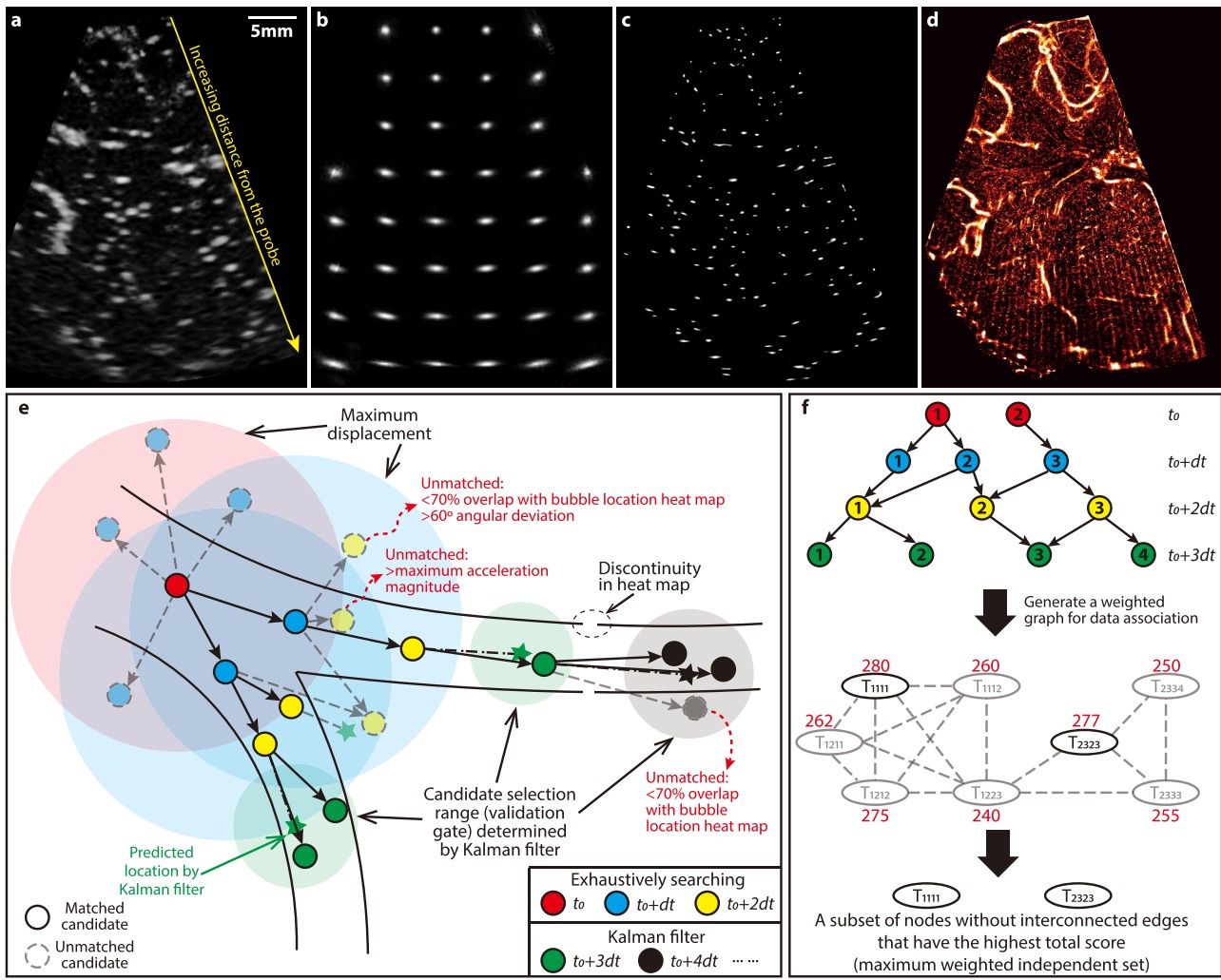

**Fig. 6 Image preprocessing and bubble tracking procedures. a** A sample raw CEUS image for piglet #0121 at baseline ICP showing original data where the bubble traces become increasingly elongated with increasing distance from the probe. One measurement was made for this case, yielding 5760 images. **b** The spatial distribution of the point spread function estimated by the blind deconvolution procedure[31]. **c** The enhanced image after deconvolution and modified histogram equalization. **d** A heatmap of the centers of all the detected bubbles, which is used for restricting the likely locations of blood vessels during the bubble tracking process. **e** The bubble tracking process highlighting several strategies used for detecting likely trajectories. Exposures are color coded as indicated in the legend. The track is initialized based on exhaustive search within a prescribed maximum displacement range for the first three exposures. The corresponding fourth and subsequent exposures are detected using a Kalman filter. Further restrictions are based on the overlap with the heatmap, bubble image morphology, and variations in directions of velocity and acceleration. **f** Illustrations of strategies for data association to determine the optimal non-overlapping trajectories: (i) generating a weighted graph by assigning each possible trajectory to a node with a score representing its likelihood of being the correct trajectory, and linking nodes that share the same bubble, and (ii) solving for the maximum weighted independent set.

frame (blue), and then repeated to identify all possible matches in the 3rd frame (yellow). A series of criteria were used for removing unlikely candidates, including a minimum overlap with the vascular map (20% for speeds <1 cm/s, 70% for speeds <5 cm/s, and 90% for higher speeds), as well as maximum variations in the bubble perimeter (50%), area (50%), flow direction (60°), and velocity magnitude (one third of the maximum prescribed distance). The overlap threshold was selected by examining numerous tracks and other thresholds were based on prior experience[34]. The remaining combinations of three exposures were considered as possible trajectories. For each track, a Kalman filter was utilized to predict the state (i.e., location and morphology) of bubbles in the fourth frame (stars in Fig. 6e). This filter was widely used for object tracking[65], including ULM[66–68]. Its prediction incorporated information of initially predicted states, correction using matched candidates falling within the uncertainty threshold, and measurement covariances. The uncertainty range was based on the Mahalanobis distance ($D_M$)[69], which expressed the difference between measured and predicted states in terms of standard deviations. Since $D_M^2$ followed a chi-squared distribution[70], the threshold for selecting a subset of the candidates (Fig. 6e green circles) was chosen by the inverse chi-squared cumulative distribution at a level of 0.95[71]. The procedure assigned

a score to each track, which was incrementally updated based on $D_M^2$ and the covariances in each time step[72].

The purpose of the next step was to select the most likely trajectory for each time step, considering that in some cases the same bubble was assigned to multiple tracks, as illustrated at the top of Fig. 6f. Starting from e.g. $t_0$, a data association procedure was carried out by solving the maximum weighted independent set (MWIS) problem[73] to ensure that trajectories did not coincide. The bubbles at each frame were numerically coded, and each potential track was labeled as $T_{ijmn}$, where $i$ corresponded to the $i$th bubbles detected at $t_0$, $j$ to a bubble detected at $t_0 + dt$, $m$ at $t_0 + 2dt$, and $n$ at $t_0 + 3dt$. Then, an undirected graph was generated by assigning each potential trajectory as a node and linking nodes if they shared any bubbles at any frame (middle panel in Fig. 6f). Each node was assigned a weight equal to the sum of the tracking scores for the recent three steps (red numbers in Fig. 6f). Since the correct subset of nodes should not be interlinked, solving for the MWIS[73] of this graph determined all the independent trajectories with the highest global tracking scores. In the shown example, the solution identified $T_{1111}$ and $T_{2323}$ as the global optimal tracks. Once this procedure was completed for the first four exposures, the Kalman filter followed by the MWIS based selection were applied to detect the bubble in the following

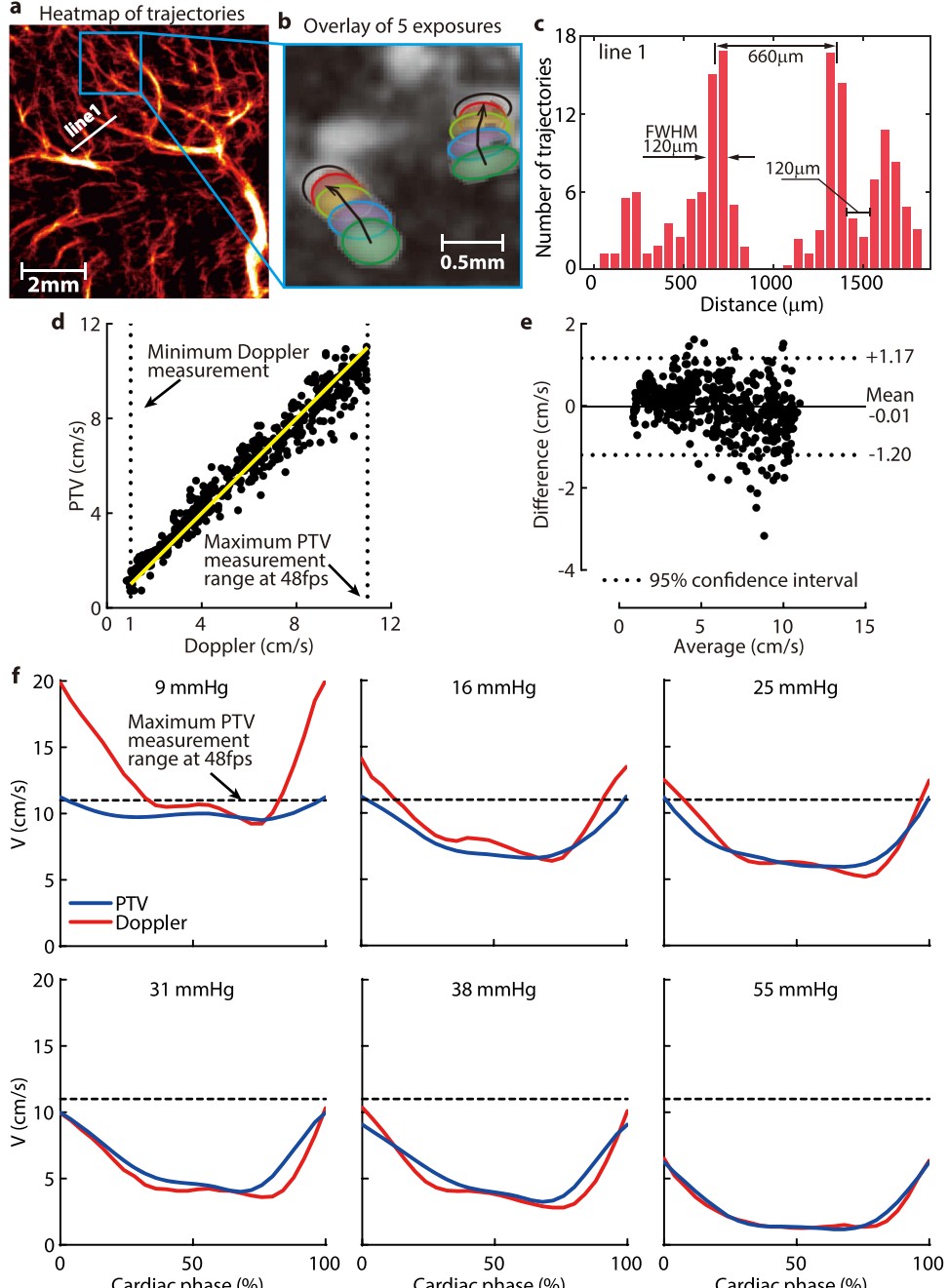

**Fig. 7 Resolution and validation of the bubble tracks. a** A sample magnified subsection of the thalamus in Fig. 1a generated from 5760 images. **b** A superposition of five exposures in the highlighted area in (**a**) showing two trajectories oriented in different directions. **c** A histogram of the number of trajectories along line 1 shown in (**a**), demonstrating, based on the full width at half maximum (FWHM), that blood micro-vessels separated by 120 mm can be readily distinguished at the current magnification. **d** A scatter plot comparing the phase-averaged velocities measured by PTV and pulsed-wave Doppler ultrasound. Results beyond the maximum PTV measurement range (11 cm/s), and below the minimum range of Doppler measurements (1 cm/s), are not included. Most of the micro-vessel data fall below the Doppler limit. **e** A Bland-Altman plot comparing the PTV and Doppler measurements, with the mean value and the 95% confidence interval highlighted. **f** A comparison of the PTV- and Doppler-measured phase-averaged velocity distributions in macro vessel #5 at varying ICP for piglet #0306. The agreement below the maximum PTV range is evident. Source data for (**d**–**f**) are provided as a Source Data file.

exposures. Finally, as an added measure to assure reliable tracking, after processing the first 33% of images, the results were used for identifying a reference flow direction. Then, the whole analysis was initiated again, this time with an additional criterion limiting the deviation in flow direction to be <15° from that of the reference. The codes for image processing and bubble tracking were developed and validated in-house while utilizing the available image processing and statistical analysis toolbox functions in MATLAB (The MathWorks, USA).

**Resolution and validation of the bubble tracks**. For all micro-vessels, only trajectories consisting of at least four exposures were used for subsequent data analysis. For the macro vessels, where the flow direction was known, three-exposure trajectories were also included in order to extend the range of maximum detectable speed, which was limited by the framerate of the present CEUS system. Figure 7b is a sample superposition of five exposures along with the corresponding bubble trajectories in the highlighted subsection of the heatmap in Fig. 7a. The current procedure could readily track bubbles with different flow directions located close to

each other. In contrast to the heatmap of bubble centers in Fig. 6d, the heatmaps visualizing the vascular structures presented in Figs. 1, 2, 4, and 7a were generated by super-positioning the bubble trajectories. To demonstrate the spatial resolution of the present measurements, Fig. 7c shows a sample heatmap profile across the arbitrarily selected line 1 marked in Fig. 7a. Here, the indicated distance between the full width at half maximum of neighboring blood vessels is 120 μm, which could be readily distinguished.

To evaluate the maximum velocity that could be measured using the present tracking procedure, results obtained for macro blood vessel #5 were compared to those measured by pulsed-wave Doppler Ultrasound at the same location. As reported[74], the uncertainty of Doppler measurement in the 2.5–10 cm/s range was lower than 5%, hence it could be used for validating the PTV data. Since the measurements were performed at different times, but under the same physiological conditions, the comparison was based on phase-averaged velocities in the cardiac cycle. Such a comparison could only be performed for the macro-vessels since the velocity in the micro-vessels falls below the detection limit of the Doppler system. The upper threshold of PTV measurements was selected as 11 cm/s by examining multiple images and assessing the maximum velocity for which three exposures could be readily detected based on the current framerate. The results are summarized in Fig. 7d–f. The scatter plot comparing phase-averaged data under the same conditions (piglet number, ICP, cardiac phase) in Fig. 7d shows that the results of the two techniques agree well within their overlapping range. Results of the Bland-Altman analysis presented in Fig. 7e confirms that the bias between the PTV and Doppler is nearly zero (−0.01 cm/s), and that the 95% confidence interval is ±1.2 cm/s. Finally, a series of phase-averaged velocity profiles for piglet #0306 (Fig. 7f) for varying ICP levels demonstrate that, for velocities falling below 11 cm/s, the Doppler and PTV measurements agree well. Deviations occur when the velocity exceeds the maximum PTV limit, resulting in a downward bias in results. It should be emphasized that the velocities in the micro-vessels, the main topic of this paper, were an order of magnitude smaller than the maximum detectability limit of PTV, i.e., they could be reliably measured at the current framerate. As for the uncertainty in CMC measurements, based on analysis of noisy synthetic data[31], the uncertainty in bubble localization was 1.5 pixels. To estimate the resulting uncertainty of CMC, for each bubble trajectory, random values varying between −1.5 to 1.5 pixels were added to the horizontal and vertical velocity components at each time point. The CMC was recalculated, and the procedure was repeated for 500 times for each case. The resulting root mean squared error in CMC was considered as the measurement uncertainty. The uncertainty in the $CMC_{cort}$ (Fig. S11) is 5% for ICP levels of 10, 20, 30, and 40 mmHg, and up to 6–7% for ICP larger than 50 mmHg.

**Statistical analysis**. The Pearson's correlation coefficients ($r$) and the corresponding two-tailed $P$ values were used to assess the correlation between the ICP and a series of parameters, including the CMC, $V_5$, and combinations of them with hemodynamic parameters. The nonlinear least square regression analysis was used to determine the functional relationships between the ICP and perfusion parameters. The agreement between the fitted functions and the measured properties were characterized by the corresponding coefficients of determination, $R^2$. Specific values of $r$, $P$, and $R^2$ were provided for each plot. The differences between two groups are tested using unpaired Welch's t-test. The statistical differences among several groups were evaluated using a 2 Way ANOVA (Analysis of Variance) Test with Benjamini-Hochberg correction for multi-comparison. The Bland-Altman analysis was used for evaluating the differences between the CEUS-PTV and the pulsed-wave Doppler ultrasound measurements. All the statistical analyses were performed using Graphpad Prism (V8.3.0, GraphPad Software, USA), as well as MATLAB 2018b (Mathworks, Cambridge, MA, USA).

**Reporting summary**. Further information on research design is available in the Nature Research Reporting Summary linked to this article.

## Data availability

The data discussed in the paper are available either in the main text or as Supplementary Material. Source data used in the figures are also provided with this paper. The data in the circulation maps, several GB, are available upon reasonable request to the corresponding author, in accordance with the guidelines of the institutions involved (Johns Hopkins University and Children's Hospital of Philadelphia). The size of the raw CEUS image files exceeds 1 TB, i.e., they are too large to be readily shared online, but could be copied & sent upon reasonable request. Source data are provided with this paper.

## Code availability

Codes for the bubble tracking are available from the corresponding author upon reasonable request.

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

## Acknowledgements

M.H. acknowledges support from NIH grants R01-NS119473 and KL2-TR001879; J.K. and Z.Z. acknowledge support from NIH grants R01-NS119473, and the Johns Hopkins University.

## Author contributions

Z.Z. and M.H. contributed equally. J.K., M.H., and T.K. conceived the project. M.H., A.S., and Z.Z. performed CEUS imaging. T.K. helped with animal management during the experiment. Z.Z. and J.K. contributed to the development of PTV code and other data analysis codes. Z.Z., J.K., and M.H. performed data analysis. All authors, i.e., Z.Z., J.K., M.H., T.K., and A.S. contributed to the manuscript preparation and revision.

## Competing interests

The authors delare no competing interests.
