## [Peer Review File · Nature Communications]

Cerebral Microcirculation Mapped by Echo Particle Tracking Velocimetry Quantifies the Intracranial Pressure and Detects IschemiaREVIEWERS' COMMENTS

Reviewer #1 (Remarks to the Author):

Thanks for the responses to my comments and the new data&analysis which have answered/clarified the points raised. A couple of more points:

1. The following claim by the authors is incorrect:

"It should be noted that all the other applications of ULM involve non-clinical high speed ultrasound systems acquiring data at framerates more than 500 frames/s..."

The very first few studies (referenced by the authors) including those from the Bochum group 2011 IUS and London group (2013 PMB and 2015 IEEE TMI), and the later study reported in the Nature Communication paper 2018 by the Bochum group all used low frame rate CEUS pre-clinical/clinical scanners as far as I know. These should be clarified.

2. A more appropriate review of the past development in relevant fields and clarification of the different terminology is required. Echo-particle tracking velocimetry from the authors' field is not the same as ULM (also known as super-resolution ultrasound). In the first two journal papers demonstrating the technology (both published in PMB 2013 by the London and Paris groups) they did not involve any tracking. Just localization can generate a super-resolution map. The tracking was separately done by the Bochum group in 2011 and then integrated into super-resolution ultrasound in 2015 by the London group. There are still advanced methods that generate localisation map but without tracking. I understand why the authors want to introduce the ETV but it must be properly put in context. Also note that in the ultrasound community the technique was not initially called ULM in the references 32-34 quoted by the authors.

Reviewer #2 (Remarks to the Author):

This manuscript presents a novel methodology to detect elevated intracranial pressure (ICP) and cerebral ischemia using a Contrast Enhanced Ultrasound (CEUS). This could be very relevant for making decisions about the early surgical treatments of elevated ICP in infants with hydrocephalus, but it may also find the application in other brain conditions in infants such as TBI. The authors performed a battery of measurements in the hydrocephalic pediatric porcine model and demonstrated that volume-averaged microvascular flow velocity (which excludes large vessels), named cerebral microcirculation (CMC) parameter, when combined with other hemodynamic parameters, correlates very highly with the ICP and has high potential to predict elevated ICP and hypoxia. I believe that authors provided adequate and detailed responses to the previously raised very relevant questions. I do not have further questions except a few comments that could be addressed in the discussion.

Overall, while presented correlations are convincing, the sample size is relatively small and there is a risk that developed data processing approaches may not work so well when applied to heterogeneous population of subjects. Microvascular heterogeneity (both structural and functional) is formidable, and it changes rapidly over first months, especially if babies are born prematurely. When coupled with the additional brain or other physiological conditions that may coexist with the hydrocephalus, it may complicate significantly how to best process and interpret CEUS data. This is of course not the reason to prematurely abandon this approach. In contrary, presented work demonstrates very high potential of this technology to contribute significantly to monitoring the state of microcirculation and brain tissue health in infants, which is very significant. As such, it should be pursued further vigorously, but it may be good to better discuss the challenges that lay ahead.

The obtained CEUS data is much richer than what is included into CMC parameter. While CMC is fairly simple observable averaged over substantial tissue region, which may help with the robustness, it may also potentially fail in presence of systematic microvascular differences between the subjects of different age and with different brain conditions. It looks plausible that in the

future, data processing algorithms could be developed that will much more take into account microvascular parameters such as vessel size, branching orders, vessel types, individual microvascular velocities, etc. This is the path that includes long learning curve but at the end, in combination with CMC and hemodynamic parameters as presented in this work, it may lead to more robust predictions. The technology is uniquely offering this possibility. I believe that manuscript will benefit from this discussion.

Finally, very minor comment is that Diffused Correlation Spectroscopy has been developed in recent years to address similar questions about microcirculation in infants and it may be appropriate to include the citations in the introduction.

1 We would like to thank the referees for reviewing our revised manuscript carefully and for making
2 constructive suggestions on how to improve it. Below is a list containing all the comments, followed
3 by our point-by-point responses to them, and how we have addressed them in the revised paper. In
4 this response document, the referees' comments are written in italics, our responses in regular script,
5 and modifications to the manuscript in blue. In the revised manuscript, the newly added/modified
6 content is highlighted, as required.

7
8 **Reviewer #1:**

9
10 **Comment #1:**

11 *Thanks for the responses to my comments and the new data & analysis which have answered/clarified*
12 *the points raised. A couple of more points:*

13
14 *1. The following claim by the authors is incorrect:*

15
16 *"It should be noted that all the other applications of ULM involve non-clinical high speed ultrasound*
17 *systems acquiring data at framerates more than 500 frames/s..."*

18
19 *The very first few studies (referenced by the authors) including those from the Bochum group 2011*
20 *IUS and London group (2013 PMB and 2015 IEEE TMI), and the later study reported in the Nature*
21 *Communication paper 2018 by the Bochum group all used low frame rate CEUS pre-clinical/clinical*
22 *scanners as far as I know. These should be clarified.*

23 **Response:** Our statement was made in the response to the reviewer but not in the manuscript, and
24 as the reviewer indicated, it was inaccurate by not acknowledging prior works using pre-clinical/clinical
25 devices. The missing information has been added to revised introduction. The following sentence has
26 been added (revised line 86-88): "Early applications have used pre-clinical/clinical CEUS scanners
27 with framerates typically less than 100 frames per second^{29,35,36,42}, and recent studies have already
28 utilized ultrafast ultrasound imaging, extending to more than 1000 frames per second^{38,41}."

29
30 **Comment #2:**

31 *2. A more appropriate review of the past development in relevant fields and clarification of the different*
32 *terminology is required. Echo-particle tracking velocimetry from the authors' field is not the same as*
33 *ULM (also known as super-resolution ultrasound). In the first two journal papers demonstrating the*
34 *technology (both published in PMB 2013 by the London and Paris groups) they did not involve any*
35 *tracking. Just localization can generate a super-resolution map. The tracking was separately done by*
36 *the Bochum group in 2011 and then integrated into super-resolution ultrasound in 2015 by the London*
37 *group. There are still advanced methods that generate localization map but without tracking. I*
38 *understand why the authors want to introduce the ETV but it must be properly put in context. Also*
39 *note that in the ultrasound community the technique was not initially called ULM in the references 32-*
40 *34 quoted by the authors.*

41 **Response:** To address the referee's concerns about the past development and the terminology, we
42 have separated the echo-PTV from the introduction of the initial works of super-resolution ultrasound
43 and ULM, and carefully revised the timeline of the development of this technique. The relevant
44 paragraph now read (revised lines: 75-89): "...In the ultrasound community, Ultrasound Localization
45 Microscopy (ULM), which is also referred to as super-resolution ultrasound, has been developed to
46 generate high spatial resolution maps of blood vessels using many localized bubble positions^{28,29}.
47 Separately, several postprocessing techniques have been developed in the fluids community to

48 quantify the flow parameters based on CEUS images, including ultrasound imaging velocimetry or
49 echo-PIV³⁰⁻³², and echo-particle tracking velocimetry^{33,34}. Combining bubble tracking with super
50 resolution ultrasound imaging has enabled mapping of the flow velocity in blood vessels³⁵⁻³⁷.
51 Pioneering *in vivo* works include the demonstration of isolated microbubble tracing in tumor vessels
52³⁵, and the super-resolution flow mapping in a mouse ear³⁶. Subsequently, this method has been
53 used for reconstructing the cerebral and renal vascular systems in rodent models³⁸⁻⁴⁰, as well as for
54 detecting a small deep-seated human cerebral aneurysm⁴¹. Early applications have used pre-
55 clinical/clinical CEUS scanners with framerates typically less than 100 frames per second^{29,35,36,42},
56 and recent studies have already utilized ultrafast ultrasound imaging, extending to more than 1000
57 frames per second^{38,41}. Further details can be found in a recent review article⁴³. ...”

58

59

60 **Reviewer #2:**

61

62 **Comment #1:**

63 *This manuscript presents a novel methodology to detect elevated intracranial pressure (ICP) and*
64 *cerebral ischemia using a Contrast Enhanced Ultrasound (CEUS). This could be very relevant for*
65 *making decisions about the early surgical treatments of elevated ICP in infants with hydrocephalus,*
66 *but it may also find the application in other brain conditions in infants such as TBI. The authors*
67 *performed a battery of measurements in the hydrocephalic pediatric porcine model and demonstrated*
68 *that volume-averaged microvascular flow velocity (which excludes large vessels), named cerebral*
69 *microcirculation (CMC) parameter, when combined with other hemodynamic parameters, correlates*
70 *very highly with the ICP and has high potential to predict elevated ICP and hypoxia. I believe that*
71 *authors provided adequate and detailed responses to the previously raised very relevant questions. I*
72 *do not have further questions except a few comments that could be addressed in the discussion.*

73 **Response:** We very much appreciate the positive comment by the referee. We have addressed each
74 of the comments in the revised discussion, as described in detail below.

75

76 **Comment #2:**

77 *Overall, while presented correlations are convincing, the sample size is relatively small and there is a*
78 *risk that developed data processing approaches may not work so well when applied to heterogeneous*
79 *population of subjects. Microvascular heterogeneity (both structural and functional) is formidable, and*
80 *it changes rapidly over first months, especially if babies are born prematurely. When coupled with the*
81 *additional brain or other physiological conditions that may coexist with the hydrocephalus, it may*
82 *complicate significantly how to best process and interpret CEUS data. This is of course not the reason*
83 *to prematurely abandon this approach. In contrary, presented work demonstrates very high potential*
84 *of this technology to contribute significantly to monitoring the state of microcirculation and brain tissue*
85 *health in infants, which is very significant. As such, it should be pursued further vigorously, but it may*
86 *be good to better discuss the challenges that lay ahead.*

87 **Response:** Indeed, the current work is based on an acute hydrocephalic pediatric porcine model, the
88 sample size is small, and we do not address complicating challenges associated with heterogeneous
89 population of subjects. Considerable work still needs to be performed to comprehensively investigate
90 the impact of physiological conditions that may complicate the interpretation of CEUS data. To
91 address the reviewer's comment, we have expanded the discussion about the sample size and the
92 potential challenges that lay ahead. In the revised version, the added text is integrated with the original
93 discussions about chronic changes to the brain perfusion caused by hydrocephalus. The revised
94 text (revised lines 373-396) now reads:

95 "Fourth, while the present findings demonstrate the potential role of cerebral micro-vessels as imaging
96 biomarkers of ICP and brain ischemia, our study is limited to a small sample size and an acute
97 hydrocephalus model. The cerebral micro-circulation and CMC vs. ICP relationship are likely to be
98 affected by hemodynamics, medications, duration and severity of prior insult, maturation, genetics,
99 metabolomics, and systemic factors. They might also be complicated by the phase of hydrocephalus,
100 i.e., whether it is acute, subacute, chronic, or acute on chronic. For example, previous studies have
101 reported contradictory results about the differences between acute and chronic phases in
102 hydrocephalus. Rashid et al.⁵⁰ report that the mean cerebral capillary diameters of rats are 4.4 ± 0.7 ,
103 4.5 ± 0.8 , and $4.7\pm 1.0\mu\text{m}$ for the acute (5-7days), chronic (3-5weeks), and control animals, respectively.
104 The corresponding mean blood flow rates are 0.6 ± 0.6 , 0.8 ± 0.7 , and $0.8\pm 0.9\text{nL/min}$, being higher
105 during the chronic phase than that of the acute phase. In contrast, in Higashi et al.⁶¹, the regional
106 CBFs in the cortex, thalamus, and midbrain show no differences between acute (~1week) and chronic
107 (4-6 weeks) phases in cat models, but both are lower than the control cases. In order to make our
108 results generalizable to patients with hydrocephalus of various etiologies and disease duration, this
109 study needs to be expanded significantly to investigate how the biomarker accuracy is affected by

110 individual variabilities, accounting for all the aforementioned factors. Moreover, taking advantage of
111 the high spatial resolution features of ULM results, future data analysis could probe into the micro-
112 vascular structures well beyond the CMC parameter. For example, one could account for the impact
113 of physiological conditions on several microvascular parameters, such as vessel sizes, branching
114 orders, vessel types, individual microvascular velocities, etc. Accounting for these parameters is likely
115 to improve the robustness of diagnostic potential of CEUS imaging in general, and assessment of ICP
116 or ischemia in particular.”

117

118 **Comment #3:**

119 *The obtained CEUS data is much richer than what is included into CMC parameter. While CMC is*
120 *fairly simple observable averaged over substantial tissue region, which may help with the robustness,*
121 *it may also potentially fail in presence of systematic microvascular differences between the subjects*
122 *of different age and with different brain conditions. It looks plausible that in the future, data processing*
123 *algorithms could be developed that will much more take into account microvascular parameters such*
124 *as vessel size, branching orders, vessel types, individual microvascular velocities, etc. This is the*
125 *path that includes long learning curve but at the end, in combination with CMC and hemodynamic*
126 *parameters as presented in this work, it may lead to more robust predictions. The technology is*
127 *uniquely offering this possibility. I believe that manuscript will benefit from this discussion.*

128 **Response:** We totally agree with the referee’s comment that the CEUS data provide much more
129 information than the CMC, and that an analysis based on the CMC is only a simple starting point for
130 a series of comprehensive follow up studies. To obtain robust predictions, these systematic studies
131 should account for the microvascular parameters under heterogeneous physiological conditions as
132 mentioned by the reviewer, hence involve more sophisticated data analysis procedures. We have
133 already incorporated the response to this comment in the revised paragraph included in the response
134 to the previous comment. Specifically, we have stated that (revised lines 390-396):

135 “Moreover, taking advantage of the high spatial resolution features of ULM results, future data
136 analysis could probe into the micro-vascular structures well beyond the CMC parameter. For example,
137 one could account for the impact of physiological conditions on several microvascular parameters,
138 such as vessel sizes, branching orders, vessel types, individual microvascular velocities, etc.
139 Accounting for these parameters is likely to improve the robustness of diagnostic potential of CEUS
140 imaging in general, and assessment of ICP or ischemia in particular.”

141

142 **Comment #4:**

143 *Finally, very minor comment is that Diffused Correlation Spectroscopy has been developed in recent*
144 *years to address similar questions about microcirculation in infants and it may be appropriate to*
145 *include the citations in the introduction.*

146 **Response:** We have added a reference to this technique in the revised introduction, and now
147 mentioned it together with other noninvasive methods. The revised text (revised lines 52-54) reads:
148 “...considerable efforts have been invested in developing noninvasive methods ¹¹. They include
149 transcranial Doppler ¹², MRI ¹³, near-infrared spectroscopy (NIRS) ¹⁴, diffuse correlation spectroscopy
150 ¹⁵, and venous ophthalmodynamometry ¹⁶.”

151

152